# OPTIMAL ATTACKS ON REINFORCEMENT LEARNING POLICIES

## ABSTRACT

Control policies, trained using the Deep Reinforcement Learning, have been recently shown to be vulnerable to adversarial attacks introducing even very small perturbations to the policy input. The attacks proposed so far have been designed using heuristics, and build on existing adversarial example crafting techniques used to dupe classifiers in supervised learning. In contrast, this paper investigates the problem of devising *optimal* attacks, depending on a well-defined attacker's objective, e.g., to minimize the main agent average reward. When the policy and the system dynamics, as well as rewards, are known to the attacker, a scenario referred to as a white-box attack, designing optimal attacks amounts to solving a Markov Decision Process. For what we call black-box attacks, where neither the policy nor the system is known, optimal attacks can be trained using Reinforcement Learning techniques. Through numerical experiments, we demonstrate the efficiency of our attacks compared to existing attacks (usually based on Gradient methods). We further quantify the potential impact of attacks and establish its connection to the smoothness of the policy under attack. Smooth policies are naturally less prone to attacks (this explains why Lipschitz policies, with respect to the state, are more resilient). Finally, we show that from the main agent perspective, the system uncertainties and the attacker can be modelled as a Partially Observable Markov Decision Process. We actually demonstrate that using Reinforcement Learning techniques tailored to POMDP (e.g. using Recurrent Neural Networks) leads to more resilient policies.

## 1 INTRODUCTION

Advances in Deep Reinforcement Learning (RL) have made it possible to train end-to-end policies achieving superhuman performance on a large variety of tasks, such as playing Atari games Mnih et al. (2013; 2016), playing Go Silver et al. (2016; 2017), as well as controlling systems with continuous state and action spaces Lillicrap et al. (2016); Schulman et al. (2015); Levine et al. (2016). Recently, some of these policies have been shown to be vulnerable to adversarial attacks at test time see e.g. Huang et al. (2017); Pattanaik et al. (2018). Even if these attacks only introduce small perturbations to the successive inputs of the policies, they can significantly impact the average collected reward by the agent.

So far, attacks on RL policies have been designed using heuristic techniques, based on gradient methods, essentially the Fast Gradient Sign Method (FGSM). As such, they do not explicitly aim at minimizing the reward collected by the agent. In contrast, in this paper, we investigate the problem of casting *optimal* attacks with well-defined objectives, for example minimizing the average reward collected by the agent. We believe that casting optimal attacks is crucial when assessing the robustness of RL policies, since ideally, the agent should learn and apply policies that resist *any* possible attack (of course with limited and reasonable amplitude).

For a given policy learnt by the agent, we show that the problem of devising an optimal attack can be formulated as a Markov Decision Process (MDP), defined through the system dynamics, the agent's reward function and policy. We are mainly interested in *black-box* attacks: to devise them, the adversary only observes the variables from which she builds her reward. For example, if the objective is to lead the system to a certain set of (bad) states, the adversary may just observe the states as the system evolves. If the goal is to minimize the cumulative reward collected by the agent, the

adversary may just observe the system states and the instantaneous rewards collected by the agent. In black-box attacks, the aforementioned MDP is unknown, and optimal attacks can be trained using RL algorithms. The action selected by the adversary in this MDP corresponds to a perturbed state to be fed to the agent's policy. As a consequence, the action space is typically very large. To deal with this issue, our optimal attack is trained using DDPG Lillicrap et al. (2016), a Deep RL algorithm initially tailored to continuous action space.

We train and evaluate optimal attacks for some OpenAI Gym Brockman et al. (2016) environments with discrete action spaces (discrete MountainCar, Cartpole, and Pong), and continuous state-action spaces (continuous MountainCar and continuous LunarLander). As expected, optimal attacks outperform existing attacks.

We discuss how the methods used to train RL policies impact their resilience to attacks. We show that the damages caused by attacks are upper bounded by a quantity directly related to the smoothness of the policy under attack[1]. Hence, training methods such as DDPG leading to smooth policies should be more robust. Finally, we remark that when under attack, the agent faces an environment with uncertain state feedback, and her sequential decision problem can be modelled as a Partially Observable MDP (POMDP). This suggests that training methods specifically tailored to POMDP (e.g. DRQN Hausknecht & Stone (2015)) result in more resistant policies. These observations are confirmed by our experiments.

## 2 RELATED WORK

Adversarial attacks on RL policies have received some attention over the last couple of years. As for now, attacks concern mainly the system at test time, where the agent has trained a policy using some Deep RL algorithm (except for Behzadan & Munir (2017), which considers an attack during training). The design of these attacks Huang et al. (2017); Pattanaik et al. (2018); Lin et al. (2017) are inspired by the Fast Gradient Sign Method (FGSM) Goodfellow et al. (2015), which was originally used to fool Deep Learning classifiers. Fast Gradient Methods (FGMs) are gradient methods that changes the input with the aim of optimizing some objective function. This function, and its gradient, are sampled using observations: in case of an agent trained using Deep RL, this function can be hence related to a distribution of actions given a state (the actor-network outputs a distribution over actions), or the $Q$-value function (the critic outputs approximate $Q$-values). In Huang et al. (2017), FGM is used to minimize the probability of the main agent selecting the optimal action according to the agent. On the other hand, in Pattanaik et al. (2018), the authors develop a gradient method that maximizes the probability of taking the action with minimal $Q$-value. The aforementioned attacks are based on the *unperturbed* policy of the agent, or its $Q$-value function, and do not consider optimizing the attack over all possible trajectories. Therefore, they can be considered a first-order approximation of an optimal attack. If the adversary wishes to minimize the average cumulative reward of the agent, she needs to work on the *perturbed* agent policy (to assess the impact of an attack, we need to compute the rewards after the attack).

Proactive and reactive defence methods have been proposed to make RL policies robust to gradient attacks. Proactive methods are essentially similar to those in supervised learning, i.e., they are based on injecting adversarial inputs during training Kos & Song (2017); Pattanaik et al. (2018). A few reactive methods have also been proposed Lin et al. (2017). There, the idea is to train a separate network predicting the next state. This network is then used to replace the state input by the predicted state if this input is believed to be too corrupted. But as we show in the paper, adversarial examples introduce partial observability, and the problem is not Markov anymore. It is known Singh et al. (1994) that deterministic memory-less policies are inadequate to deal with partial observability, and some kind of memory is needed to find a robust policy. In this work, we make use of recurrent neural networks to show that we can use techniques from the literature of Partially Observable MDPs to robustify RL policies.

It is finally worth mentioning that there has been some work studying the problem of robust RL with a game-theoretical perspective Morimoto & Doya (2005); Pinto et al. (2017). However, these papers

---

[1]A policy is *smooth* when the action it selects smoothly varies with the state. Refer to proposition 5.1 for details.

consider a very different setting where the adversary has a direct impact on the system (not indirectly through the agent as in our work). In a sense, they are more related to RL for multi-agent systems.

## 3 PRELIMINARIES

This section provides a brief technical background on Markov Decision Process and Deep Reinforcement Learning. The notions and methods introduced here are used extensively in the remainder of the paper.

**Markov Decision Process.** A Markov Decision Process (MDP) defines a discrete-time controlled Markov chain. It is defined by a tuple $\mathcal{M} = \langle \mathcal{S}, (\mathcal{A}_s, s \in \mathcal{S}), P, p_0, q \rangle$. $\mathcal{S}$ is the finite state space, $\mathcal{A}_s$ is the finite set of actions available in state $s$, $P$ represents the system dynamics where $P(\cdot|s,a)$ is the distribution of the next state given that the current state is $s$ and the selected action is $a$, $p_0$ is the distribution of the initial state, and $q$ characterizes rewards where $q(\cdot|s,a)$ is the distribution of the collected reward in state is $s$ when action $a$ is selected. We denote by $r(s,a)$ the expectation of this reward. We assume that the average reward is bounded: for all $(s,a)$, $|r(s,a)| \leq R$. A policy $\pi$ for $\mathcal{M}$ maps the state and to a distribution of the selected action. For $a \in \mathcal{A}_s$, $\pi(a|s)$ denotes the probability of choosing action $a$ in state $s$. The value function $V^\pi$ of a policy $\pi$ defines for every $s$ the average cumulative discounted reward under $\pi$ when the system starts in state $s$: $V^\pi(s) = \mathbb{E}[\sum_{t \geq 0} \gamma^t r(s_t^\pi, a_t^\pi)|s_0 = s]$ where $s_t^\pi$ and $a_t^\pi$ are the state and the selected action at time $t$ under policy $\pi$, and $\gamma \in (0,1)$ is the discount factor. The $Q$-value function $Q^\pi$ of policy $\pi$ maps a (state, action) pair $(s,a)$ to the average cumulative discounted reward obtained starting in state $s$ when the first selected action is $a$ and subsequent actions are chosen according to $\pi$: $Q^\pi(s,a) = r(s,a) + \gamma \mathbb{E}_{s' \sim P(\cdot|s,a)}[V^\pi(s')]$. For a given MDP $\mathcal{M}$, the objective is to devise a policy with maximal value in every state.

**Deep Reinforcement Learning** To train policies with maximal rewards for MPDs with large state or action spaces, Deep Reinforcement Learning consists in parametrizing the set of policies or some particular functions of interest such as the value function or the $Q$-value function of a given policy by a deep neural network. In this paper, we consider various Deep RL techniques to train either the policy of the agent or the attack. These include:

**DQN** Mnih et al. (2013): the network aims at approximating the $Q$-function of the optimal policy.

**DDPG** Lillicrap et al. (2016): an actor-critic method developed to deal with continuous state-action spaces. To this aim, the actor returns a single action rather than a distribution over actions.

**DRQN** Hausknecht & Stone (2015): introduces *memory* in DQN by replacing the (post-convolutional) layer by a recurrent LSTM. The use of this recurrent structure allows us to deal with partial observability, and DRQN works better with POMDPs.

## 4 OPTIMAL ADVERSARIAL ATTACKS

To attack a given policy $\pi$ trained by the main agent, an adversary can slightly modify the sequence of inputs of this policy. The changes are imposed to be small, according to some distance, so that the attack remains difficult to detect. The adversary aims at optimizing inputs with a precise objective in mind, for example, to cause as much damage as possible to the agent.

We denote by $\mathcal{M} = \langle \mathcal{S}, (\mathcal{A}_s, s \in \mathcal{S}), P, p_0, q \rangle$ the MDP solved by the agent, i.e., the agent policy $\pi$ has been trained for $\mathcal{M}$.

### 4.1 THE ATTACK MDP

To attack the policy $\pi$ the adversary proceeds as follows. At time $t$, the adversary observes the system state $s_t$. She then selects a perturbed state $\bar{s}_t$, which becomes the input of the agent policy $\pi$. The agent hence chooses an action according to $\pi(\cdot|\bar{s}_t)$. The adversary successively collects a random reward defined depending on her objective, which is assumed to be a function of the true state and the action selected by the agent. An attack is defined by a mapping $\phi : \mathcal{S} \to \mathcal{S}$ where $\bar{s} = \phi(s)$ is the perturbed state given that the true state is $s$.

**Constrained perturbations.** For the attack to be imperceptible, the input should be only slightly modified. Formally, we assume that we can define a notion of distance $d(s, s')$ between two states $s, s' \in \mathcal{S}$. The state space of most RL problems can be seen as a subset of a Euclidean space, in which case $d$ is just the Euclidean distance. We impose that given a state $s$ of the system, the adversary can only select a perturbed state $\bar{s}$ in $\bar{\mathcal{A}}_s^\epsilon = \{\bar{s} \in \mathcal{S} : d(s, \bar{s}) \leq \epsilon\}$. $\epsilon$ is the maximal amplitude of the state perturbation.

**System dynamics and agent's reward under attack.** Given that the state is $s_t = s$ at time $t$ and that the adversary selects a modified state $\bar{s} \in \bar{\mathcal{A}}_s^\epsilon$, the agent selects an action according to the distribution $\pi(\cdot|\bar{s})$. Thus, the system state evolves to a random state with distribution $\bar{P}^\pi(\cdot|s, \bar{s}) := \sum_a \pi(a|\bar{s}) P(\cdot|s, a)$. The agent instantaneous reward is a random variable with distribution $\sum_a \pi(a|\bar{s}) q(\cdot|s, a)$.

**Adversary's reward.** The attack is shaped depending on the adversary's objective. The adversary typically defines her reward as a function of the true state and the action selected by the agent, or more concisely as a direct function of the reward collected by the agent. The adversary might be interested in guiding the agent to some specific states. In control systems, the adversary may wish to induce oscillations in the system output or to reduce its controllability (this can be realized by choosing a reward equal to the energy spent by the agent, i.e., proportional to $a^2$ if the agent selects $a$). The most natural objective for the adversary is to minimize the average cumulative reward collected by the agent. In this case, the adversary would set her instantaneous reward equal to the opposite of the agent's reward.

We denote by $\bar{q}^\pi(\cdot|s, \bar{s})$ the distribution of the reward collected by the adversary in state $s$ when she modifies the input to $\bar{s}$, and by $\bar{r}^\pi(s, \bar{s})$ its expectation. For example, when the adversary wishes to minimize the agent's average cumulative reward, we have $\bar{r}^\pi(s, \bar{s}) = -\sum_a \pi(a|\bar{s}) r(s, a)$.

We have shown that designing an optimal attack corresponds to identifying a policy $\phi$ that solves the following MDP: $\bar{\mathcal{M}}^\pi = \langle \mathcal{S}, (\bar{\mathcal{A}}_s^\epsilon, s \in \mathcal{S}), \bar{P}^\pi, p_0, \bar{q}^\pi \rangle$. When the parameters of this MDP are known to the adversary, a scenario referred to as *white box* attack, finding the optimal attack accounts to solving the MDP, e.g., by using classical methods (value or policy iteration). More realistically, the adversary may ignore the parameters of $\bar{\mathcal{M}}$, and only observe the state evolution and her successive instantaneous rewards. This scenario is called *black-box* attack, and in this case, the adversary can identify the optimal attack using Reinforcement Learning algorithms.

For ease of notation, we will denote the value of the adversarial policy $\phi$ as $V^\phi$ (for the MDP $\bar{\mathcal{M}}^\pi$), even though it depends on $\pi$. Finally, observe that when the adversarial attack policy is $\phi$, then the system dynamics and rewards correspond to a scenario where the main agent applies the perturbed policy $\pi \circ \phi$, which is defined by the distributions $(\pi \circ \phi)(\cdot|s) := \mathbb{E}_{\bar{s} \sim \phi(s)}[\pi(\cdot|\bar{s})], s \in \mathcal{S}$. Hence, we can denote the value of the perturbed policy as $V^{\pi \circ \phi}$ (for the MDP $\mathcal{M}$).

### 4.2 Minimizing agent's average reward

Next, we give interesting properties of the MDP $\bar{\mathcal{M}}^\pi$. When the adversary aims at minimizing the average cumulative reward of the agent, then the reward collected by the agent can be considered as a cost by the adversary. In this case, $\bar{r}^\pi(s, \bar{s}) = -\sum_a \pi(a|\bar{s}) r(s, a)$, and one can easily relate the value function of an attack policy $\phi$ to that of the agent policy $\pi \circ \phi$:

$$V^\phi(s) = -V^{\pi \circ \phi}(s), \quad \forall s \in \mathcal{S}. \tag{1}$$

The $Q$—value function of an attack policy $\phi$ can also be related to the $Q$—value function of the agent policy under attack $\pi \circ \phi$. Indeed: for all $s, \bar{s} \in \mathcal{S}$.

$$Q^\phi(s, \bar{s}) = -\bar{r}^\pi(s, \bar{s}) + \gamma \mathbb{E}_{s' \sim \bar{P}^\pi(\cdot|s, \bar{s})}[V^\phi(s')],$$

$$= -\mathbb{E}_{a \sim \pi(\cdot|\bar{s})}\Big[r(s, a) + \gamma \mathbb{E}_{s' \sim P(\cdot|s, a)}[V^{\pi \circ \phi}(s')]\Big] = -\mathbb{E}_{a \sim \pi(\cdot|\bar{s})}[Q^{\pi \circ \phi}(s, a)].$$

In particular, if the agent policy is deterministic ($\pi(s) \in \mathcal{A}_s$ is the action selected under $\pi$ in state $s$), we simply have:

$$Q^\phi(s, \bar{s}) = -Q^{\pi \circ \phi}(s, \pi(\bar{s})), \quad \forall s, \bar{s} \in \mathcal{S}. \tag{2}$$

Equation (2) has an important consequence when we train an attack policy using Deep Learning algorithms. It implies that evaluating the $Q$-value of an attack policy $\phi$ can be done by evaluating

the $Q$-value function of the perturbed agent policy $\pi \circ \phi$. In the case of off-policy learning, the former evaluation would require to store in the replay buffer experiences of the type $(s, \bar{s}, r)$ ($r$ is the observed reward), whereas the latter just stores $(s, \pi(\bar{s}), r)$ (but it requires to observe the actions of the agent). This simplifies considerably when the state space is much larger than the action space.

## 4.3 TRAINING OPTIMAL ATTACKS

Training an optimal attack can be very complex, since it corresponds to solving an MDP where the action space size is similar to that of the state space. In our experiments, we use two (potentially combined) techniques to deal with this issue: (i) feature extraction when this is at all possible, and (ii) specific Deep RL techniques tailored to very large (or even continuous) state and action spaces, such as DDPG.

**Feature extraction.** One may extract important features of the system admitting a low-dimensional representation. Formally, this means that by using some expert knowledge about the system, one can identify a bijective mapping between the state $s$ in $\mathcal{S}$, and its features $z$ in $\mathcal{Z}$ of lower dimension. In general, this mapping can also be chosen such that its image of the ball $\mathcal{A}_s^\epsilon$ is easy to represent in $\mathcal{Z}$ (typically this image would also be a ball around $z$, the feature representation of $s$). It is then enough to work in $\mathcal{Z}$. When $\mathcal{Z}$ is of reasonable size, one may then rely on existing value-based techniques (e.g. DQN) to train the attack. An example where features can be easily extracted is the Atari game of pong: the system state is just represented by the positions of the ball and the two bars. Finally, we assume to be very likely that an adversary trying to disrupt the agent policy would conduct an advanced feature engineering work before casting her attack.

**Deep RL: DDPG.** In many systems, it may not be easy to extract interesting features. In that case, one may rely on Deep RL algorithms to train the attack. The main issue here is the size of the action space. This size prevents us to use DQN (that would require to build a network with as many outputs as the number of possible actions), but also policy-gradient and actor-critic methods that parametrize randomized policies such as TRPO (indeed we can simply not maintain distributions over actions). This leads us to use DDPG, an actor-critic method that parametrizes deterministic policies. In DDPG, we consider policies parametrized by $\omega$ ($\phi_\omega$ is the policy parametrized by $\omega$), and its approximated $Q$—value function $Q_\theta$ parametrized by $\theta$. We also maintain two corresponding target networks. The objective function to maximize is $J(\omega) = \mathbb{E}_{s \sim p_0}[V^{\phi_\omega}(s)]$. $\phi$ is updated using the following gradient Silver et al. (2014):

$$\nabla_\omega J(\omega) = \mathbb{E}_{s \sim \rho^{\phi^e}}[\nabla_{\bar{s}} Q_\theta(s, \phi_\omega(s)) \nabla_\omega \phi_\omega(s)], \tag{3}$$

where $\phi^e$ corresponds to $\phi_\omega$ with additional exploration noise, and $\rho^{\phi^e}$ denotes the discounted state visitation distribution under the policy $\phi^e$. In the case where the adversary's objective is to minimize the average reward of the agent, in view of (2), the above gradient becomes:

$$\nabla_\omega J(\omega) = -\mathbb{E}_{s \sim \rho^{\phi^e}}[\nabla_a Q_\theta(s, \pi(\phi_\omega(s))) \nabla_{\bar{s}} \pi(\phi_\omega(s)) \nabla_\omega \phi_\omega(s)]. \tag{4}$$

Finally, to ensure that the selected perturbation computed by $\phi_\omega$ belongs to $\mathcal{A}_s^\epsilon$, we add a last fixed layer to the network that projects onto the ball centred in 0, with radius $\epsilon$. This is done by applying the function $x \mapsto \min\left(1, \frac{\epsilon}{\|x\|}\right) x$ to $x + e$, where $x$ is the output before the projection layer, and $e$ is the exploration noise. After this projection, we add the state $s$ to get $\phi^e(s)$.

The pseudo-code of the algorithm is provided in Algorithm 1 in the Appendix. Algorithm 2 presents a version of the algorithm using the gradient (4). A diagram of the actor and critic networks are also provided in Figure 8 of the Appendix.

**Gradient-based exploration.** In case the adversary knows the policy $\pi$ of the main agent, we can leverage this knowledge to tune the exploration noise $e$. Similarly, to gradient methods Huang et al. (2017); Pattanaik et al. (2018), we suggest using the quantity $\nabla_s J(\pi(\cdot|s), y)$ to improve the exploration process , where $J$ is the cross-entropy loss between $\pi(\cdot|s)$ and a one-hot vector $y$ that encodes the action with minimum probability in state $s$. This allows exploring directions that might minimize the value of the unperturbed policy, which boosts the rate at which the optimal attack is learnt. At time $t$, the exploration noise could be set to $e_t'$, a convex combination of a white exploration noise $e_t$ and $f(s_t) = g(-\nabla_s J(\pi(\cdot|s_t), y_t))$, where $g$ is a normalizing function. Following this gradient introduces bias, hence we randomly choose when to use $e_t$ or $e_t'$ (more details can be found in the Appendix).

## 5 ROBUSTNESS OF POLICIES

In this section, we briefly discuss which training methods should lead to agent policies more resilient to attacks. We first quantify the maximal impact of an attack on the cumulative reward of the agent policy, and show that it is connected to the smoothness of the agent policy. This connection indicates that the smoother the policy is, the more resilient it is to attacks. Next, we show that from the agent perspective, an attack induces a POMDP. This suggests that if training the agent policy using Deep RL algorithms tailored to POMDP yields more resilient policies.

**Impact of an attack and policy smoothness.** When the agent policy $\pi$ is attacked using $\phi$, one may compute the cumulative reward gathered by the agent. Indeed, the system evolves as if the agent policy was $\pi \circ \phi$. The corresponding value function satisfies the Bellman equation: $V^{\pi \circ \phi}(s) = \mathbb{E}_{a \sim \pi \circ \phi(\cdot|s)}\Big[r(s,a) + \gamma \mathbb{E}_{s' \sim P(\cdot|s,a)}[V^{\pi \circ \phi}(s')]\Big]$. Starting from this observation, we derive an upper bound on the impact of an attack (see the Appendix for a proof):

**Proposition 5.1.** *For any $s \in \mathcal{S}$, let[2] $\alpha_{\pi,\varepsilon}(s) = \max_{\bar{s} \in \mathcal{A}_s^\varepsilon} \|\pi(\cdot|s) - \pi(\cdot|\bar{s})\|_{TV}$. We have:*

$$\|V^\pi - V^{\pi \circ \phi}\|_\infty \leq 2 \frac{\|\alpha_{\pi,\varepsilon}\|_\infty}{1 - \gamma} \left(R + \gamma \|V^\pi\|_\infty\right). \tag{5}$$

*Assume now that $\pi$ is smooth in the sense that, for all $s, s' \in \mathcal{S}$, $\|\pi(\cdot|s) - \pi(\cdot|s')\|_{TV} \leq Ld(s,s')$, then*

$$\|V^\pi - V^{\pi \circ \phi}\|_\infty \leq 2 \frac{L\varepsilon}{1 - \gamma} \left(R + \gamma \|V^\pi\|_\infty\right). \tag{6}$$

In the above, $\alpha_{\pi,\varepsilon}(s)$ quantifies the potential impact of applying a feasible perturbation to the state $s$ on the distribution of actions selected by the agent in this state (note that it decreases to 0 as $\varepsilon$ goes to 0). The proposition establishes the intuitive fact that when the agent policy is smooth (i.e., varies smoothly with the input), it should be more resilient to attacks. Indeed, in our experiments, policies trained using Deep RL methods known to lead to smooth policies (e.g. DDPG for RL problems with continuous state and action spaces) resist better to attacks.

**Induced POMDP and DRQN.** When the agent policy is under the attack $\phi$, the agent perceives the true state $s$ only through its perturbation $\phi(s)$. More precisely, the agent observes a perturbed state $\bar{s}$ only, with probability $\mathcal{O}(\bar{s}|s) = \mathbb{1}_{\{\phi(s)=\bar{s}\}}$ (in general $\mathcal{O}$ could also be a distribution over states if the attack is randomized). Therefore, an optimall perturbation policy $\phi$ induces a deterministic POMDP. It is known Spaan (2012); Astrom (1965); Singh et al. (1994) that solving POMDPs requires "memory", i.e., policies whose decisions depend on the past observations (non-Markovian). Hence, we expect that training the agent policy using Deep RL algorithms tailored to POMDP produces more resilient policies. In Hausknecht & Stone (2015) they empirically demonstrate that recurrent controllers have a certain degree of robustness against missing information, even when trained with full state information. We use this to pro-actively extract robust features, not biased by a specific adversary policy $\phi$, as in adversarial training. This is confirmed in our experiments, comparing the impact of attacks on policies trained by DRQN to those trained using Deep RL algorithms without recurrent structure (without memory).

## 6 EXPERIMENTAL EVALUATION

We evaluate our attacks on four OpenAI Gym Brockman et al. (2016) environments (A: discrete MountainCar, B: Cartpole, C: continuous MountainCar, and D: continuous LunarLander), and on E: Pong, an Atari 2600 game in the Arcade Learning Environment Bellemare et al. (2013). In Appendix, we also present the results of our attacks for the toy example of the Grid World problem, to illustrate why gradient-based attack are sub-optimal.

**Agent's policies.** The agent policies are trained using DQN or DRQN in case of discrete action spaces (environments A, B, and E), and using DDPG when the action space is continuous (environments C and D). In each environment and for each training algorithm, we have obtained from 3 policies for DQN, 1 for DRQN (achieving at least 95% of the performance of the best policy reported in OpenAI Gym).

---

[2]$\|\cdot\|_{TV}$ is the Total Variation distance, and for any $V \in \mathbb{R}^{|\mathcal{S}|}$, $\|V\|_\infty = \max_{s \in \mathcal{S}} |V(s)|$.

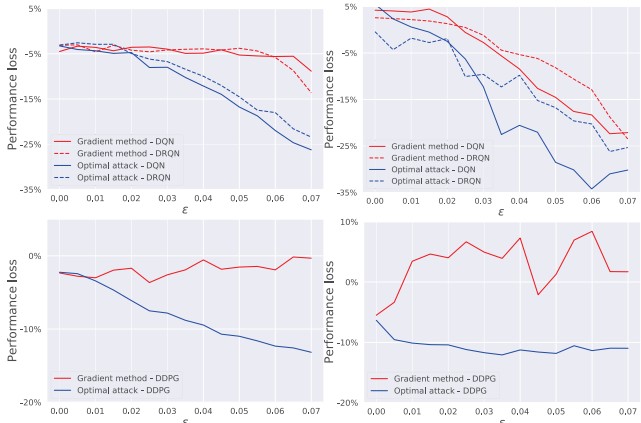

Figure 1: Average performance loss vs. attack amplitude $\varepsilon$: (top-left) discrete MountainCar, (top-right) Cartpole, (bottom-left) continuous MountainCar, (bottom-right) LunarLander. The attack policy $\phi$ has been trained only for $\varepsilon = 0.05$ In red is shown the FGM attack, in blue the attack proposed in this paper.

**Adversarial attack.** For each environment A, B, C, and D, we train one adversarial attack using DDPG against one of the agent's policies, and for a perturbation constraint $\varepsilon = 0.05$ (we use the same normalization method as in Huang et al. (2017)). For results presented below, we use uniformly distributed noise; results with gradient-based noise are discussed in the Appendix. To get attacks for other values of $\varepsilon$, we do not retrain our attack but only change the projection layer in our network. For environment E, we represent the state as a 4-dimensional feature vector, and train the attack using DQN in the feature space. The full experimental setup is presented in the Appendix.

### 6.1 OPTIMAL VS. GRADIENT-BASED ATTACKS

We compare our attacks to the FGM attacks (gradient-based) proposed in Pattanaik et al. (2018) (we found that these are the most efficient). To test a particular attack, we run it against the 3 agent's policies, using 10 random seeds (e.g. involved in the state initialization), and 30 episodes. We hence generate up to 1800 episodes, and average the cumulative reward of the agent.

In Figure 1, we plot the performance loss for different the perturbation amplitudes $\varepsilon$ in the 4 environments. Observe that the optimal attack consistently outperforms the gradient-based attack, and significantly impact the performance of the agent. Also note that since we have trained our attack for $\varepsilon = 0.05$, it seems that this attack generalizes well for various values of $\varepsilon$. In Appendix, we present similar plots but not averaged over the 3 agent's policies, and we do not see any real difference — suggesting that attacks generalize well to different agent's policies.

From Figure 1, for environments with discrete action spaces (the top two sets of curves), the resilience of policies trained by DRQN is confirmed: these policies are harder to attack.

Policies trained using DDPG for environments with continuous action spaces (the bottom two sets of curves in Figure 1) are more difficult to perturb than those trained in environments with discrete action spaces. The gradient-based attack does not seem to be efficient at all, at least in the range of attack amplitude $\varepsilon$ considered. In the case of LunarLander some state components were not normalized, which may explain why a bigger value of $\varepsilon$ is needed in order to see a major performance loss (at least $\varepsilon = 0.1$). Our optimal attack performs better, but the impact of the attack is not as significant as in environments with discrete action spaces: For instance, for discrete and continuous MountainCar, our attack with $\varepsilon = 0.07$ decreases the performance of the agent by 30% and 13%, respectively.

### 6.2 ATTACK ON PONG, BASED ON FEATURE EXTRACTION

We consider the game of Pong where the state is an 84x84 pixel image. The agent's policies were trained using DQN using as inputs the successive images. As for the attack, we extracted a 4-dimensional feature vector fully representing the state: this vector encodes the position of the centre

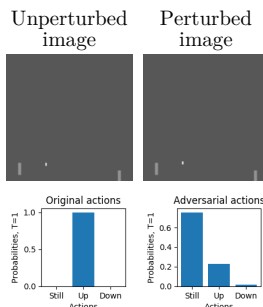 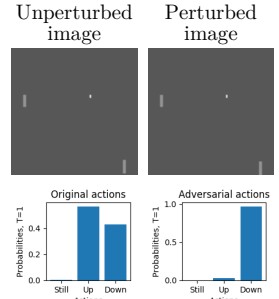 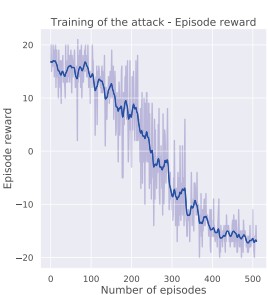

Figure 2: Game of Pong. (Left part) Images and corresponding agent's action distributions before and after the attack. (Right part) Average agent's reward at different phases of the attack training.

of mass of the three relevant objects in the image (the ball and the two bars). Extracting the features is done using a high pass filter to detect the contours. We need 4-dimensional vectors since the two bars have a single degree of freedom, and the ball has two.

We limit the amplitude of the attack in the feature space directly. More precisely, we impose that the adversary just changes one of the 4 components of the feature vector by one pixel only. For example, the adversary can move up one bar by one pixel. We found that the amplitude of the resulting attack in the true state space (full images) corresponds to $\varepsilon = 0.013$.

The attack is trained using DQN in the feature space. In Figure 2 (left part), we present a few true and perturbed images, and below the corresponding distributions of the actions selected by the agent without and with the attack (the probabilities are calculated from the output of the network using a softmax function with temperature $T = 1$). Moving objects by one pixel can perturb considerably the agent's policy. In Figure 2 (right part), we show the evolution of the performance of the agent's policy under our attack during the training of the latter. Initially, the agent's policy achieves the maximum score, but after 500 episodes of training, the agent's score has become close to the lowest possible score in this game (-20). This score can be reached in an even fewer number of episodes when tuning the training parameters. The gradient-based attacks performed in Huang et al. (2017) cannot be directly compared to ours, since these attacks change 4 frames at every time step, while ours modifies only the last observation. They also modify the values of every pixel in frames (using 32-bit precision over the reals [0,1]), whereas we change only the values of just a few (using 8-bit precision over the integers [0,255]).

## 7 CONCLUSION

In this paper we have formulated the problem of devising an optimal black-box attack that does not require access to the underlying policy of the main agent. Previous attacks, such as FGM Huang et al. (2017), make use of white-box assumptions, i.e., knowing the action-value function Q, or the policy, of the main agent. In our formulation, we do not assume this knowledge. Deriving an optimal attack is important in order to understand how to build RL policies robust to adversarial perturbations.

The problem has been formulated as a Reinforcement Learning problem, where the goal of the adversary is encapsulated in the adversarial reward function. The problem becomes intractable when we step out of toy problems: we propose a variation of DDPG to compute the optimal attack. In the white-box case, instead of using FGM, we propose to use a gradient-based method to improve exploration.

Adapting to such attacks requires solving a Partially Observable MDP, hence we have to resort to non-markovian policies Singh et al. (1994). It can be achieved by adopting recurrent layers, as in DRQN Hausknecht & Stone (2015). We also show that Lipschitz policies have desirable robustness properties.

We validated our algorithm on different environments. In all cases we have found out that our attack outperforms gradient methods. This is more evident in discrete action spaces, whilst in continuous spaces it is more difficult to perturb for small values of $\varepsilon$, which may be explained by the bound we provide for Lipschitz policies. Finally, policies that use memory seem to be more robust in general.

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

# A APPENDIX

## A.1 OTHER EXPERIMENTAL RESULTS

**Gridworld.** We use as a toy-example a $6 \times 6$ gridworld, with $\ell_1$ distance, and perturbation magnitude $\varepsilon = 1.0$. The goal of the game is to reach the top-left or bottom-right corner in the fewest number of steps, with reward $-1$ for each step spent in the game. We implemented both the gradient methods inspired by Huang et al. Huang et al. (2017) and Pattanaik et al. Pattanaik et al. (2018), and compared them with our proposed method to find the optimal perturbation policy $\phi$. The implementation of those algorithms is shown in Algorithm 3, 4. In figure 3 is shown the undiscounted value of the various attacks: we can see the value of the worst attack defined in Pattanaik et al. (2018) does not achieve the same values that are achieved from the optimal perturbation $\phi$.

|  | $-\infty$ | $-\infty$ | $-\infty$ | $-\infty$ | $-\infty$ |
|---|---|---|---|---|---|
| -1 | $-\infty$ | $-\infty$ | $-\infty$ | $-\infty$ | $-\infty$ |
| -2 | $-\infty$ | $-\infty$ | $-\infty$ | $-\infty$ | $-\infty$ |
| -3 | $-\infty$ | $-\infty$ | $-\infty$ | $-\infty$ | $-\infty$ |
| -4 | $-\infty$ | $-\infty$ | $-\infty$ | $-\infty$ | $-\infty$ |
| $-\infty$ | $-\infty$ | -3 | -2 | -1 | |

|  | $-\infty$ | $-\infty$ | $-\infty$ | $-\infty$ | $-\infty$ |
|---|---|---|---|---|---|
| -1 | -2 | -3 | -4 | $-\infty$ | $-\infty$ |
| -2 | -3 | -4 | -5 | -4 | $-\infty$ |
| -3 | -4 | -7 | -6 | -3 | $-\infty$ |
| -4 | -9 | -8 | -3 | -2 | $-\infty$ |
| -11 | -10 | -3 | -2 | -1 | |

|  | $-\infty$ | $-\infty$ | $-\infty$ | $-\infty$ | $-\infty$ |
|---|---|---|---|---|---|
| -1 | $-\infty$ | $-\infty$ | $-\infty$ | $-\infty$ | $-\infty$ |
| -2 | $-\infty$ | $-\infty$ | $-\infty$ | $-\infty$ | $-\infty$ |
| -3 | $-\infty$ | $-\infty$ | $-\infty$ | $-\infty$ | $-\infty$ |
| -4 | $-\infty$ | $-\infty$ | $-\infty$ | $-\infty$ | $-\infty$ |
| -5 | $-\infty$ | -3 | -2 | -1 | |

Figure 3: Un-discounted value of the perturbed policy of the main agent. Left: our proposed attack, middle: attack in Huang et al. (2017), right: attack in Pattanaik et al. (2018).

**OpenAI Gym Environments.** Figure 4 presents the results for the discrete MountainCar environment A. There, we plot the distribution of the performance loss (in %) due to the attack for $\varepsilon = 0.05$, when the agent policies have been trained using DQN (top) or DRQN (bottom).

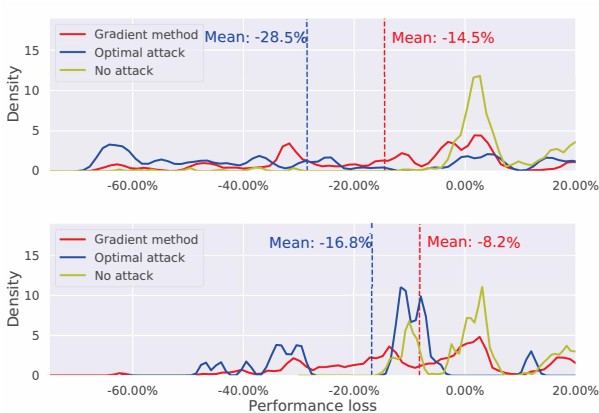

Figure 4: Performance loss distribution under the optimal and gradient-based attacks with amplitude $\varepsilon = 0.05$ for the discrete MountainCar environment. The attacks are against policies trained using DQN (top) and DRQN (bottom).

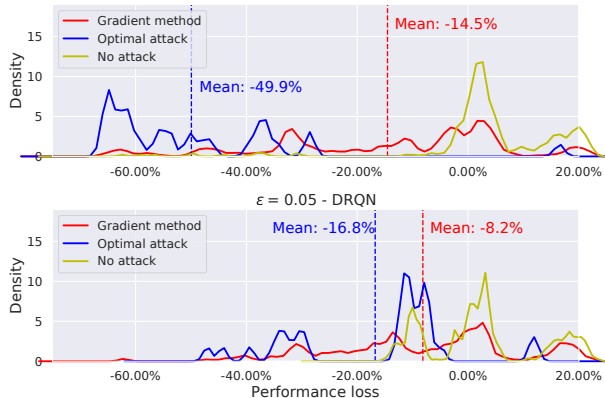

Figure 5: Mountaincar environment - distribution of the average discounted reward performance decrease for $\varepsilon = 0.05$, with respect to the main agent's policy the attacker has trained for.

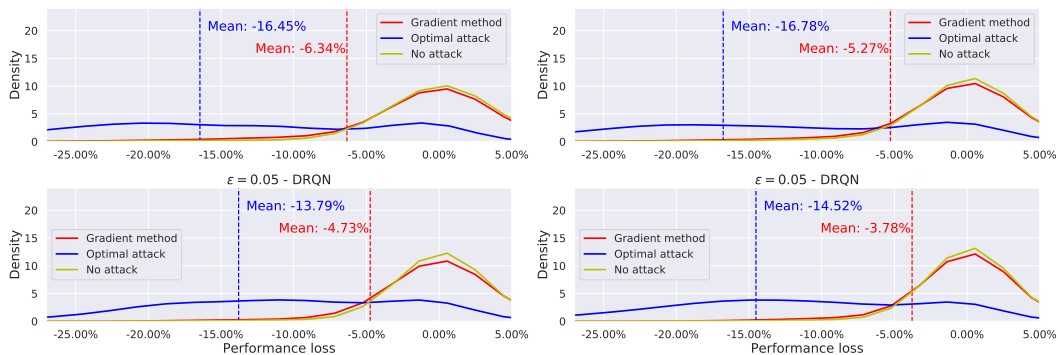

Figure 6: Cartpole environment - distribution of the average discounted reward performance decrease for $\varepsilon = 0.05$. On the left we show the distribution with respect to the main agent's policy the attacker was trained on, and on the right an average across all the main agent's policies.

In Figure 5-8 we show the performance loss distribution for $\varepsilon = 0.05$, and in Figure 9 the average performance loss for different values of $\varepsilon$, all computed when the adversary attacks the agent's policy it was trained for. Additionally, for environment B, C, D, we show the loss distribution averaged across all the main agent's policies.

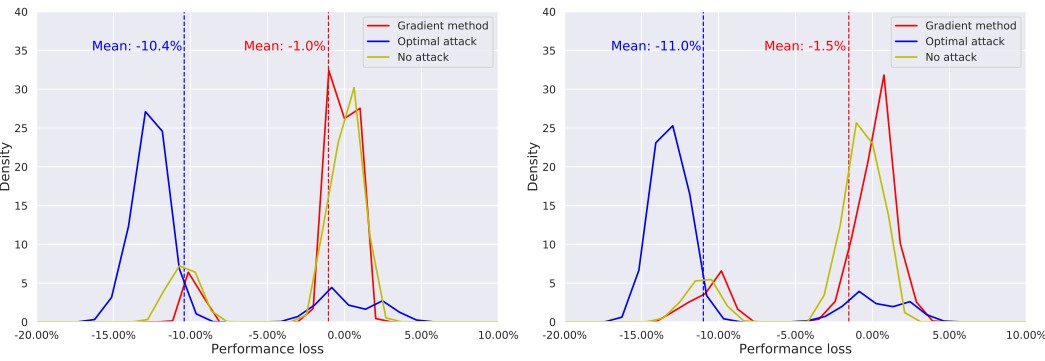

Figure 7: Continuous MountainCar environment - distribution of the average discounted reward performance decrease for $\varepsilon = 0.05$. On the left we show the distribution with respect to the main agent's policy the attacker was trained on, and on the right an average across all the main agent's policies.

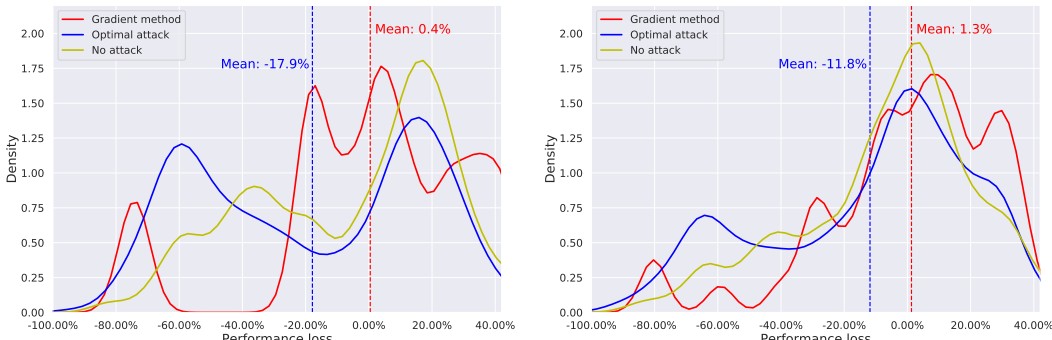

Figure 8: LunarLander environment - distribution of the average discounted reward performance decrease for $\varepsilon = 0.05$. On the left we show the distribution with respect to the main agent's policy the attacker was trained on, and on the right an average across all the main agent's policies.

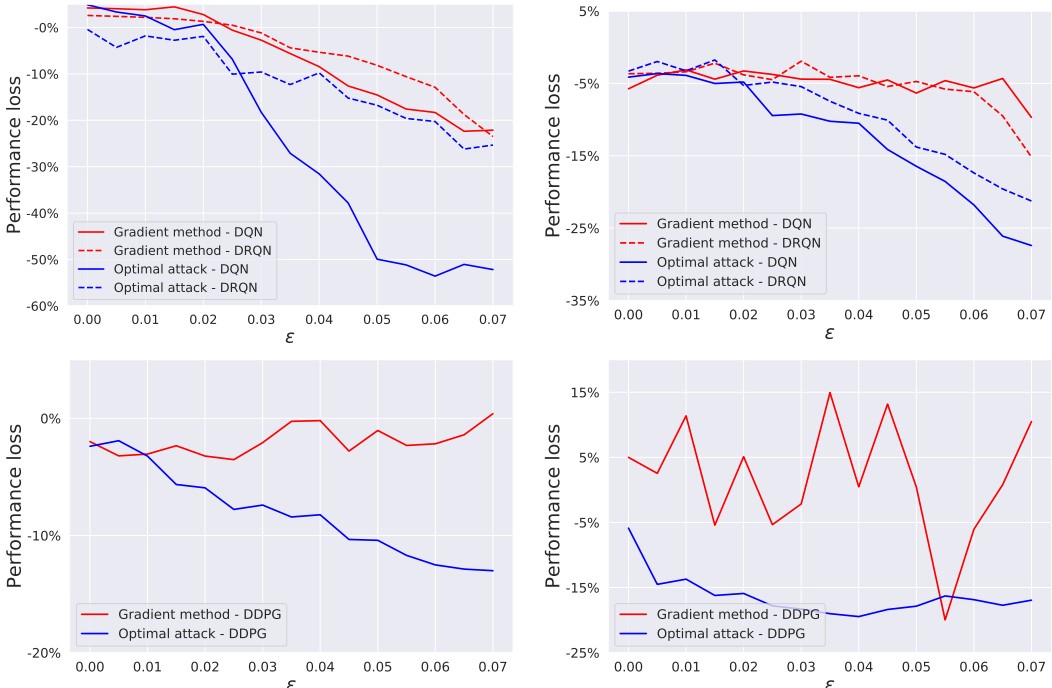

Figure 9: Average performance loss vs. attack amplitude $\varepsilon$ in the various environments, when the adversary attacks the main agent's policy it was trained on. (top-left) discrete MountainCar, (top-right) Cartpole, (bottom-left) continuous MountainCar, (bottom-right) LunarLander. Attacks are against using policies obtained using DQN, DRQN, or DDPG.

**Gradient based exploration** Gradient-based exploration tries to exploit knowledge of the unperturbed policy $\pi$ of the main agent. Caution should be taken, in order to reduce bias, and to minimize the risk of exploring like an on-policy method. If we follow the information contained in $\pi$ without restriction we would end up in a sub-optimal solution, that minimizes the value of the unperturbed policy $\pi$. The exploration noise is given by the following equation:

$$\hat{e}_t = \begin{cases} e_t, & X_t = 0, \\ (1 - \omega_t)e_t + \omega_t g(-\nabla_s J(\pi(a|s_t), y)), & X_t = 1 \end{cases} \tag{7}$$

where $g$ is a normalizing function such as $g(x) = \frac{x}{\|x\|}$, or $g(x) = \mathrm{sign}(x)$, $X_t$ is a Bernoulli random variable with $P(X_t = 1) = p$, and $\omega_t = \max_a \pi(a|s_t) - \min_a \pi(a|s_t)$. $J$ is the cross-entropy loss between $\pi(a|s_t)$ and a one-hot vector $y$ that encodes the action with minimum probability in state $s_t$.

If $\pi$ is not known, but the action-value function $Q^\pi$ is known, we can compute $\pi(\cdot|s)$ by applying the softmax function on the action-values of a particular state $s$.

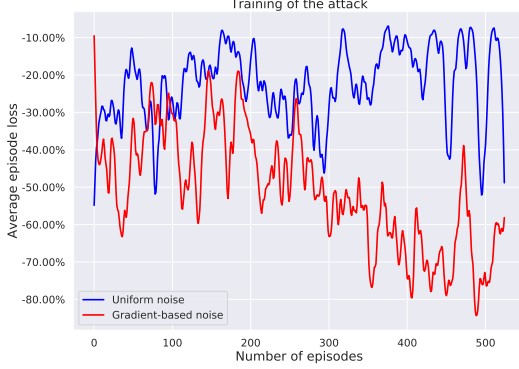

| Reward statistics | No attack | FGM | Uniform exploration | Gradient-based exploration |
|---|---|---|---|---|
| Mean | $-103.4$ | $-134.6$ | $-163.2$ | $-173.6$ |
| Std | $9.19$ | $33.12$ | $34.95$ | $29.97$ |
| Min | $-82$ | $-81$ | $-81$ | $-85$ |
| Max | $-115$ | $-200$ | $-200$ | $-200$ |

Figure 10: Mountaincar environment. On the left is shown the average episode loss when exploring using a gradient-based exploration methods vs. uniform noise exploration. On the right are shown statistics regarding episodes reward for $\varepsilon = 0.05$, and norm $\ell_2$, after training.

The gradient of the cross-entropy loss computes the direction that increases the probability of taking the action with minimum value when the policy $\pi$ is unperturbed. The random variable $X_t$, instead, is necessary in order to reduce bias induced by following this gradient, while $\omega_t$ quantifies how strongly an agent prefers the optimal action in state $s_t$. Intuitively, in case $\pi$ is an optimal policy, the lower $\omega_t$ is, the less impact the optimal action has in that state. This guarantees that we follow the gradient of $J$ mostly when we are in critical states. In Figure 10 we show the low-pass filtered loss during the training of the attack. We clearly see an improvement in performance, although it highly depends on the selection of the parameters.

We have tested this type of exploration on Mountaincar, with $p = 0.35$, and temperature of the softmax function set to 1. The exploration noise $e_t$ is drawn from a Uniform distribution with parameters shown in Table 3 (Mountaincar). In Figure 10 we compare gradient-based exploration vs. uniform noise exploration. On the left, we show the low-pass filtered loss during the training of the attack, with $\varepsilon = 0.05$ and $\ell_2$ norm. On the right of Figure 10 we compare episodes reward statistics after training. Results in the table were averaged across 5 different seeds, with 100 episodes for each seed, for a total of 500 episodes for each method. We see an improvement in performance, although it highly depends on the selection of the parameters.

## A.2 PROOF OF PROPOSITION 5.1

*Proof.* Let

$$g^\pi(s,a) = \mathbb{E}_{s' \sim P(\cdot|s,a)}[V^\pi(s')], \quad \text{and} \quad g^{\pi \circ \phi}(s,a) = \mathbb{E}_{s' \sim P(\cdot|s,a)}[V^{\pi \circ \phi}(s')]. \quad (8)$$

We will now proceed by bounding the quantity $|(V^\pi - V^{\pi \circ \phi})(s)|$:

$$|(V^\pi - V^{\pi \circ \phi})(s)| = \left| \mathbb{E}_{a \sim \pi(\cdot|s)}[r(s,a) + \gamma g^\pi(s,a)] - \mathbb{E}_{a \sim \pi \circ \phi(\cdot|s)}[r(s,a) + \gamma g^{\pi \circ \phi}(s,a)] \right|, \quad (9)$$

$$= \left| \mathbb{E}_{\bar{s} \sim \phi(\cdot|s)}\left[ \mathbb{E}_{a \sim \pi(\cdot|s)}[r(s,a) + \gamma g^\pi(s,a)] - \mathbb{E}_{a \sim \pi(\cdot|\bar{s})}[r(s,a) + \gamma g^{\pi \circ \phi}(s,a)] \right] \right|, \quad (10)$$

where in the last line we made use of the following equality $\mathbb{E}_{a \sim \pi \circ \phi(\cdot|s)}[\cdot] = \mathbb{E}_{\bar{s} \sim \phi(\cdot|s)}[\mathbb{E}_{a \sim \pi(\cdot|\bar{s})}[\cdot]]$. Now, by making use of $|\mathbb{E}[x]| \leq \mathbb{E}[|x|]$ we get:

$$|(V^\pi - V^{\pi \circ \phi})(s)| = \left| \mathbb{E}_{\bar{s} \sim \phi(\cdot|s)}\left[ \mathbb{E}_{a \sim \pi(\cdot|s)}[r(s,a) + \gamma g^\pi(s,a)] - \mathbb{E}_{a \sim \pi(\cdot|\bar{s})}[r(s,a) + \gamma g^{\pi \circ \phi}(s,a)] \right] \right|, \quad (11)$$

$$\leq \mathbb{E}_{\bar{s} \sim \phi(\cdot|s)}\left| \left[ \mathbb{E}_{a \sim \pi(\cdot|s)}[r(s,a) + \gamma g^\pi(s,a)] - \mathbb{E}_{a \sim \pi(\cdot|\bar{s})}[r(s,a) + \gamma g^{\pi \circ \phi}(s,a)] \right] \right|. \quad (12)$$

At this point consider a random variable $X$ that can assume values only in a bounded set $K$. Let $X_\infty$ be a bound on the absolute value of $X$. Then, for any two distributions $f_1$, $f_2$ whose support is $K$ we have $|\mathbb{E}_{f_1}[X] - \mathbb{E}_{f_2}[X]| \le 2X_\infty \|f_1 - f_2\|_{TV}$:

$$|\mathbb{E}_{f_1}[X] - \mathbb{E}_{f_2}[X]| = \Big| \sum_{x \in K} x f_1(x) - \sum_{x \in K} x f_2(x)) \Big| \le X_\infty \sum_{x \in K} |f_1(x) - f_2(x)| = 2X_\infty \|f_1 - f_2\|_{TV}. \tag{13}$$

By using the previous inequality we can bound the reward difference:

$$\mathbb{E}_{\bar{s} \sim \phi(\cdot|s)} |\mathbb{E}_{a \sim \pi(\cdot|s)}[r(s,a)] - \mathbb{E}_{a \sim \pi(\cdot|\bar{s})}[r(s,a)]| \le 2R \mathbb{E}_{\bar{s} \sim \phi(\cdot|s)} \|\pi(s) - \pi(\bar{s})\|_{TV}, \tag{14}$$
$$\le 2R\alpha_{\pi,\varepsilon}(s), \tag{15}$$

where $\alpha_{\pi,\varepsilon}(s) = \max_{\bar{s} \in X_s^\varepsilon} \|\pi(s) - \pi(\bar{s})\|_{TV}$. The inequality becomes

$$|(V^\pi - V^{\pi \circ \phi})(s)| \le 2\alpha_\varepsilon(s)R + \gamma \mathbb{E}_{\bar{s} \sim \phi(\cdot|s)} \Big| \mathbb{E}_{a \sim \pi(\cdot|s)}[g^\pi(s,a)] - \mathbb{E}_{a \sim \pi(\cdot|\bar{s})}[g^{\pi \circ \phi}(s,a)] \Big|. \tag{16}$$

Now, let $g^\pi(s,a) = g^{\pi \circ \phi}(s,a) + \Delta(s,a)$, and observe that $g^\pi(s,a)$ is uniformly bounded by $V_\infty^\pi = \|V^\pi\|_\infty$, and $\Delta(s,a)$ by $\|V^\pi - V^{\pi \circ \phi}\|_\infty$, thus

$$\mathbb{E}_{\bar{s} \sim \phi(\cdot|s)} \Big| \mathbb{E}_{a \sim \pi(\cdot|s)}[g^\pi(s,a)] - \mathbb{E}_{a \sim \pi(\cdot|\bar{s})}[g^{\pi \circ \phi}(s,a)] \Big|, \tag{17}$$

$$= \mathbb{E}_{\bar{s} \sim \phi(\cdot|s)} \Big| \mathbb{E}_{a \sim \pi(\cdot|s)}[g^\pi(s,a)] - \mathbb{E}_{a \sim \pi(\cdot|\bar{s})}[g^\pi(s,a) - \Delta(s,a)] \Big|, \tag{18}$$

$$\le \mathbb{E}_{\bar{s} \sim \phi(\cdot|s)} \Big[ 2\alpha_{\pi,\varepsilon}(s)V_\infty^\pi + \|V^\pi - V^{\pi \circ \phi}\|_\infty \Big] = 2\alpha_{\pi,\varepsilon}(s)V_\infty^\pi + \|V^\pi - V^{\pi \circ \phi}\|_\infty. \tag{19}$$

Hence, we can deduce the following inequality:

$$|(V^\pi - V^{\pi \circ \phi})(s)| \le 2R\alpha_{\pi,\varepsilon}(s) + 2\gamma\alpha_{\pi,\varepsilon}(s)V_\infty^\pi + \gamma\|V^\pi - V^{\pi \circ \phi}\|_\infty, \tag{20}$$
$$= 2\alpha_{\pi,\varepsilon}(s)(R + \gamma V_\infty^\pi) + \gamma\|V^\pi - V^{\pi \circ \phi}\|_\infty. \tag{21}$$

By taking the term $\gamma\|V^\pi - V^{\pi \circ \phi}\|_\infty$ on the left hand side, dividing by $1 - \gamma$, and finally taking the supremum over $s \in \mathcal{S}$ we obtain the result:

$$\|V^\pi - V^{\pi \circ \phi}\|_\infty \le 2\frac{\|\alpha_{\pi,\varepsilon}\|_\infty}{1 - \gamma}(R + \gamma V_\infty^\pi). \tag{22}$$

Assume now the main agent's policy is smooth, such that $\|\pi(s) - \pi(s')\|_{TV} \le Ld(s,s'), L \in \mathbb{R}_{\ge 0}$. We can now bound the term $\mathbb{E}_{\bar{s} \sim \phi(\cdot|s)} |\mathbb{E}_{a \sim \pi(\cdot|s)}[r(s,a)] - \mathbb{E}_{a \sim \pi(\cdot|\bar{s})}[r(s,a)]|$ as follows:

$$\mathbb{E}_{\bar{s} \sim \phi(\cdot|s)} |\mathbb{E}_{a \sim \pi(\cdot|s)}[r(s,a)] - \mathbb{E}_{a \sim \pi(\cdot|\bar{s})}[r(s,a)]| \le \mathbb{E}_{\bar{s} \sim \phi(\cdot|s)}[2R\|\pi(s) - \pi(\bar{s})\|_{TV}], \tag{23}$$
$$\le 2LR\mathbb{E}_{\bar{s} \sim \phi(\cdot|s)}[d(s,\bar{s})], \tag{24}$$
$$= 2LR\bar{d}(s), \tag{25}$$

where $\bar{d}(s) = \mathbb{E}_{\bar{s} \sim \phi(\cdot|s)}[d(s,\bar{s})]$ denotes the average weighted distance between elements of $X_s^\varepsilon$ and $s$, according to the distribution $\phi(\cdot|s)$. Equivalently we also have

$$\mathbb{E}_{\bar{s} \sim \phi(\cdot|s)} \Big| \mathbb{E}_{a \sim \pi(\cdot|s)}[g^\pi(s,a)] - \mathbb{E}_{a \sim \pi(\cdot|\bar{s})}[g^{\pi \circ \phi}(s,a)] \Big| \le \mathbb{E}_{\bar{s} \sim \phi(\cdot|s)} \Big[ 2V_\infty^\pi \|\pi(s) - \pi(\bar{s})\|_{TV} \tag{26}$$

$$+ \|V^\pi - V^{\pi \circ \phi}\|_\infty \Big], \tag{27}$$

$$\le 2LV_\infty^\pi[\bar{d}(s) + \|V^\pi - V^{\pi \circ \phi}\|_\infty], \tag{28}$$

from which we obtain

$$|(V^\pi - V^{\pi \circ \phi})(s)| \le 2LR\bar{d}(s) + \gamma 2L\bar{d}(s)V_\infty^\pi + \gamma\|V^\pi - V^{\pi \circ \phi}\|_\infty, \tag{29}$$
$$= 2L\bar{d}(s)(R + \gamma V_\infty^\pi) + \gamma\|V^\pi - V^{\pi \circ \phi}\|_\infty, \tag{30}$$

and

$$\|V^\pi - V^{\pi \circ \phi}\|_\infty \le 2\sup_{s \in \mathcal{S}} \frac{L\bar{d}(s)}{1 - \gamma}(R + \gamma V_\infty^\pi) \le 2\frac{L\varepsilon}{1 - \gamma}(R + \gamma V_\infty^\pi). \tag{31}$$

$$\square$$

### A.2.1 Algorithms

**Attack training algorithm.** In this section we describe the DDPG algorithms discussed in Section 4.3. We will denote the projection layer and the output before the projection layer respectively by $l_\Pi$ and $x_{\phi,t}$. Algorithm 1 refers to the black-box attack, whilst Algorithm 2 makes use of $\pi$, if it is known. Apart from the projection layer, and the change in the gradient step in the white-box case, both algorithms follow the same logic as the usual DDPG algorithm, explained in Lillicrap et al. (2016). Moreover, projections have a regularization term $\lambda$ in order to prevent division by 0.

---

**Algorithm 1** Training adversarial attack

---

1: **procedure** ATTACKTRAINING($\pi, T_{\text{training}}, N_B$)
2:     Initialize critic and actor network $Q_\theta, \phi_\omega$ with weights $\theta$ and $\phi$.
3:     Initialize target critic and actor networks $Q_{\theta^-}, \phi_{\phi^-}$ with weights $\theta^- \leftarrow \theta, \phi^- \leftarrow \phi$.
4:     Initialize replay buffer $\mathcal{D}$.
5:     **for** episode 1:M **do**
6:         Initialize environment and observe state $s_0$, and set initial time $t \leftarrow 0$.
7:         **repeat**
8:             Select a perturbation and add exploration noise $\bar{s}_t \leftarrow s_t + l_\Pi(x_{\phi,t} + e_t)$.
9:             Feed perturbation $\bar{s}_t$ to main agent and observe reward and state $\bar{r}_t, s_{t+1}$.
10:             Append transition $(s_t, \bar{s}_t, \bar{r}_t, s_{t+1})$ to $\mathcal{D}$.
11:             **if** $t \equiv 0 \mod T_{\text{training}}$ **then**
12:                 Sample random minibatch $B \sim \mathcal{D}$ of $N_B$ elements.
13:                 Set target for the i-th element of $B$ to $y_i = \bar{r}_i + \gamma Q_{\theta^-}(s_{i+1}, \phi_{\phi^-}(s_{i+1}))$.
14:                 Update critic by minimizing loss $\frac{1}{N_B} \sum_{i=1}^{N_B} (y_i - Q_\theta(s_i, \bar{s}_i))^2$.
15:                 Update actor using sampled policy gradient

$$\nabla_\omega J \approx \frac{1}{N_B} \sum_{i=1}^{N_B} \nabla_{\bar{s}} Q_\theta(s_i, \bar{s}) \nabla_\omega \phi_\phi(s_i)|_{\bar{s}=\phi(s_i)}.$$

16:                 Slowly update the target networks:

$$\theta^- \leftarrow (1-\tau)\theta^- + \tau\theta, \quad \phi^- \leftarrow (1-\tau)\phi^- + \tau\phi.$$

17:             **end if**
18:             $t \leftarrow t + 1$.
19:         **until** episode is terminal.
20:     **end for**
21: **end procedure**

---

### A.2.2 Gradient-based attacks.

Gradient-based attacks can be adapted to MDPs by solving the optimization problem these methods are trying to solve. There are two main techniques that were developed, one by Huang et al. Huang et al. (2017), and the other one from Pattanaik et al. Pattanaik et al. (2018): the former tries to minimize the probability of the optimal action $a*$, whilst the latter maximizes the probability of taking the worst action, $a_*$, defined as the action that attains the minimum value for the unperturbed policy $\pi$. Pseudo-code for the former is given in Algorithm 3, while for the latter is in Algorithm 4.

### A.3 Experimental setup

**Hardware and software setup.** All experiments were executed on a stationary desktop computer, featuring an Intel Xeon Silver 4110 CPU, 48GB of RAM and a GeForce GTX 1080 graphical card. Ubuntu 18.04 was installed on the computer, together with Tensorflow 1.13.1 and CUDA 10.0.

**Deep-learning framework.** We set up our experiments within the Keras-RL framework Plappert (2016), together with OpenAI Gym Brockman et al. (2016) as interface to the test environments used in this paper.

---

**Algorithm 2** Training adversarial attack 2

---

1: **procedure** ATTACKTRAINING2($\pi, T_{\text{training}}, N_B$)
2:  Initialize critic network $Q_\theta$ and actor $\hat{\phi}_\omega$ with weights $\theta$ and $\phi$.
3:  Initialize target critic and actor networks $Q_{\theta^-}, \phi_{\phi_T}$ with weights $\theta^- \leftarrow \theta, \phi^- \leftarrow \phi$.
4:  Initialize replay buffer $\mathcal{D}$.
5:  **for** episode 1:M **do**
6:      Initialize environment and observe state $s_0$.
7:      Set initial time $t \leftarrow 0$.
8:      **repeat**
9:          Select a perturbation and add exploration noise $\bar{s}_t \leftarrow s + l_\Pi(x_{\phi,t} + e_t)$.
10:         Feed perturbation $\bar{s}_t$ to main agent and observe reward and state $\bar{r}_t, s_{t+1}$.
11:         Append transition $(s_t, a_t, \bar{r}_t, s_{t+1})$ to $\mathcal{D}$, where $a_t = \pi(\bar{s}_t)$.
12:         **if** $t \equiv 0 \mod T_{\text{training}}$ **then**
13:             Sample random minibatch $B \sim \mathcal{D}$ of $N_B$ elements.
14:             Set target for the i-th element of $B$ to $y_i = \bar{r}_i - \gamma Q_{\theta^-}(s_{i+1}, \pi(\phi_{\phi^-}(s_{i+1})))$.
15:             Update critic by minimizing loss $\frac{1}{N_B} \sum_{i=1}^{N_B}(y_i + Q_\theta(s_i, a_i))^2$.
16:             Update actor using sampled policy gradient

$$\nabla_\omega J \approx -\frac{1}{N_B} \sum_{i=1}^{N_B} \nabla_a Q_\theta(s_i, a) \nabla_{\bar{s}} \pi(\bar{s}) \nabla_\omega \phi_\phi(s_i)|_{a=\pi(\phi(s_i)), \bar{s}=\phi(s_i)}.$$

17:             Slowly update the target networks:

$$\theta^- \leftarrow (1-\tau)\theta^- + \tau\theta,$$
$$\phi^- \leftarrow (1-\tau)\phi^- + \tau\phi.$$

18:         **end if**
19:         $t \leftarrow t + 1$.
20:     **until** episode is terminal.
21:  **end for**
22: **end procedure**

---

**Algorithm 3** Adversarial Attack - Adaptation of Huang et al. (2017).

---

1: **procedure** ATTACKPROCEDURE1($s, \pi, X_s^\varepsilon$)
2:  $a^* \leftarrow \arg\max_a \pi(a|s)$                                    ▷ Best action in state $s$
3:  $\bar{s} \leftarrow s$
4:  **for all** $s' \in X_s^\varepsilon$ **do**                             ▷ Loop trough the $\varepsilon$-neighbor states
5:      **if** $\pi(a^*|s') < \pi(a^*|\bar{s})$ **then**                     ▷ Evaluate if $a^*$ is not optimal in $j$
6:          $\bar{s} \leftarrow s'$
7:      **end if**
8:  **end for**
9:  **return** $\bar{s}$
10: **end procedure**

---

**Algorithm 4** Adversarial Attack - Adaptation of Pattanaik et al. (2018).

---

1: **procedure** ATTACKPROCEDURE($s, \pi, Q^\pi, X_s^\varepsilon$)
2:  $q^* \leftarrow Q^\pi(s, \arg\max_a \pi(a|s))$
3:  $\bar{s} \leftarrow s$
4:  **for all** $s' \in X_s^\varepsilon$ **do**                             ▷ Loop trough the $\varepsilon$-neighbor states
5:      $q \leftarrow Q^\pi(s, \arg\max_a \pi(a|s'))$
6:      **if** $q < q^*$ **then**                                           ▷ Evaluate if optimal action in $s'$ is worse in $s$
7:          $q^* \leftarrow q$
8:          $\bar{s} \leftarrow s'$
9:      **end if**
10: **end for**
11:  **return** $\bar{s}$
12: **end procedure**

---

All the adversarial methods used in this paper, including Gradient attacks, and DRQN, have been implemented from scratch using the Keras-RL framework. All the code and experiments can be found at the following url https://bit.ly/2JAD7CF.

**Neural network architectures.**

Depending on the environment we use a different neural network architecture, as shown in table 1. The network architecture was not optimised, but was chosen large enough also to have a better comparison with Gradient methods since they depend on the dimensionality of the network, and may fail for small-sized networks. For all environments we set the frame skip to 4. For the adversary,

| Neural network structure | MountainCar DQN | MountainCar Actor | Q-Critic | LunarLander Actor | Q-Critic | Cartpole DQN |
|---|---|---|---|---|---|---|
| Layer 1 | FC(512, ReLU) | FC(400, ReLU) | FC(400, ReLU) | FC(400, ReLU) | FC(400, ReLU) | FC(256, ReLU) |
| Layer 2 | FC(256, ReLU) | FC(300, ReLU) | FC(300, ReLU) | FC(300, ReLU) | FC(300, ReLU) | FC(256, ReLU) |
| Layer 3 | FC(64, ReLU) | FC(1, Tanh) | FC(1) | FC(2, Tanh) | FC(1) | FC(2) |
| Layer 4 | FC(3) | - | - | - | - | - |

Table 1: Neural network structure for the policies of the main agent. FC stands for Fully Connected layer. Inside the parentheses is indicated how many units were used, and the activation function for that layer.

instead, we used a different actor-critic architecture, although we have not tried to optimise it. The model can be seen in figure 11. In the figure $\dim(S)$ denotes the dimensionality of the state space. The neural network structure for the game of Pong is the same as the one used in Huang et al. (2017), for both the adversary and the main agent.

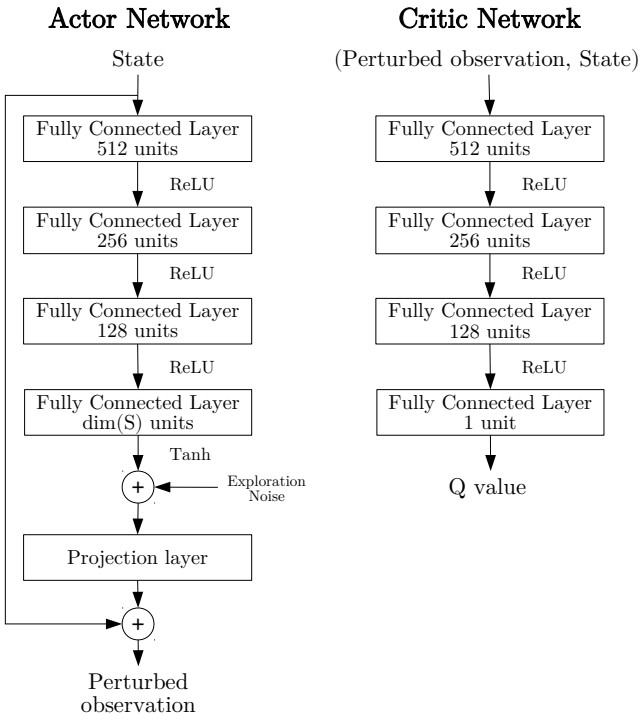

Figure 11: Diagram of the adversarial agent perturbing observations of the environment. On the right is shown the environment model for the malicious agent. The equivalent MDP is denoted by $\bar{\mathcal{M}}$.

**Data collection and preprocessing.** Every data point is an average across 10 different seeds, where for each seed we have tested an attack against all of the main agent's policies, and each test considers 30 episodes (10 in the case of FGSM for Continuous Mountaincar and LunarLander). In Table 2

is shown the number of policies for the main agent and the number of episodes collected. Density plots were preprocessed using the function seaborn.kdeplot with Gaussian kernel and bandwidth 0.05 (0.15 for LunarLander). Gradient-based exploration data was preprocessed using a first order acausal Butterworth filter with critical frequency $\omega_n = 0.1$.

| Algorithm/Environment | MountainCar | MountainCar-Continuous | Cartpole | LunarLander |
|---|---|---|---|---|
| DQN | 3 (1800) | - | 3 (1800) | - |
| DRQN | 1 (600) | - | 1 (600) | - |
| DDPG | - | 3 (1200) | - | 3 (1200) |

Table 2: Number of policies for the main agent for each Algorithm/Environment pair. In parenthesis is shown the total number of episodes taken.

**Settings for gradient-attacks.** For gradient-attacks we have always used the same norm as the one used by the optimal attack. For the Softmax layer we have always used a temperature of $T = 1$. We compute the gradient using the method from Pattanaik et al. Pattanaik et al. (2018), which has proven to be better than the one proposed by Huang et al.Huang et al. (2017). In order to speed-up the computation we do not sample the attack magnitude from a random distribution, but fix it to the maximum value $\varepsilon$.

**Training procedure and parameters.** All the training parameters for the main agent are shown in table 3. Parameters were chosen based on previous experience, so that the policy trains in a fixed number of training steps. Policies were trained until they were within 5% of the optimal value referenced in the OpenAI Gym documentation Brockman et al. (2016), across an average of the last 100 episodes. For the game of Pong we use same as Huang et al. (2017)

| Parameters/Environment | MountainCar | MountainCar | LunarLander | Cartpole | Pong |
|---|---|---|---|---|---|
| Action space | Discrete | Continuous | Continuous | Discrete | Discrete |
| Algorithm used | DQN/DRQN | DDPG | DDPG | DQN/DRQN | DQN |
| Reference value | 50.0 | 90.0 | 200.0 | 195.0 | 20 |
| Number of steps | $4 \cdot 10^4$ | $2 \cdot 10^4$ | $3 \cdot 10^4$ | $22 \cdot 10^3$ | $4 \cdot 10^6$ |
| Number of warmup steps | 100 | $10^3$ | $3 \cdot 10^3$ | $10^3$ | $5 \cdot 10^4$ |
| Replay memory size | $4 \cdot 10^4$ | $2 \cdot 10^4$ | $15 \cdot 10^3$ | $22 \cdot 10^3$ | $5 \cdot 10^5$ |
| Discount factor | 0.99 | 0.99 | 0.99 | 0.99 | 0.99 |
| Learning rate | $10^{-3}$ | $(10^{-4}, 10^{-3})$ | $(10^{-4}, 10^{-3})$ | $10^{-4}$ | $25 \cdot 10^{-4}$ |
| Optimizer | Adam | Adam | Adam | Adam | Adam |
| Gradient clipping | Not set | 1.0 | 1.0 | 1.0 | Not set |
| Exploration style | $\epsilon$-greedy | Gaussian noise | Gaussian noise | $\epsilon$-greedy | $\epsilon$-greedy |
| Initial exploration rate/ Std (DDPG) | 0.95 | 2 | 1 | 0.95 | 0.95 |
| Final exploration rate/ Std (DDPG) | 0.1 | $10^{-3}$ | $10^{-3}$ | 0.1 | 0.05 |
| Exploration steps | $28 \cdot 10^3$ | $14 \cdot 10^3$ | $21 \cdot 10^3$ | $11 \cdot 10^3$ | $35 \cdot 10^5$ |
| Batch size | 32 | 64 | 192 | 256 | 32 |
| Target model update | $10^2$ | $10^{-3}$ | $10^{-3}$ | 200 | $10^4$ |
| Initial random steps - Training | 0 | 0 | 0 | 0 | 0 |
| Initial random steps - Testing | 10 | 10 | 10 | 10 | 50 |
| Frame-skip | 4 | 4 | 4 | 4 | 4 |

Table 3: Training settings for the policies of the main agent. When using DDPG we sometimes used different parameters for the actor and the critic. In that case in the table are shown 2 parameters, the first one for the actor, the second one for the critic.

For the adversary, we used similar parameters, shown in Table 4.

| Parameters/Environment | MountainCar | MountainCar Continuous | LunarLander | Cartpole | Pong |
|---|---|---|---|---|---|
| Algorithm used | DDPG | DDPG | DDPG | DDPG | DQN+Features |
| Number of steps | $4 \cdot 10^4$ | $6 \cdot 10^4$ | $5 \cdot 10^4$ | $4 \cdot 10^4$ | $2 \cdot 10^6$ |
| Number of warmup steps | $4 \cdot 10^3$ | $3 \cdot 10^3$ | $5 \cdot 10^3$ | $4 \cdot 10^3$ | $10^4$ |
| Replay memory size | $15 \cdot 10^3$ | $3 \cdot 10^4$ | $15 \cdot 10^3$ | $15 \cdot 10^3$ | $50 \cdot 10^4$ |
| Discount factor | 0.99 | 0.99 | 0.99 | 0.99 | 0.99 |
| Learning rate | $(10^{-4}, 10^{-3})$ | $(10^{-4}, 10^{-3})$ | $(10^{-4}, 10^{-3})$ | $(10^{-3}, 10^{-2})$ | $25 \cdot 10^{-4}$ |
| Optimizer | Adam | Adam | Adam | Adam | Adam |
| Gradient clipping | 1.0 | 1.0 | 1.0 | 1.0 | Not set |
| Exploration style | Uniform noise | Uniform noise | Uniform noise | Uniform noise | $\epsilon$-greedy |
| Initial exploration settings | $[-0.5, 0.5]$ | 2 | $[-0.5, 0.5]$ | $[-0.3, 0.3]$ | 0.95 |
| Final exploration settings | $[-10^{-3}, 10^{-3}]$ | $[-10^{-3}, 10^{-3}]$ | $[-10^{-3}, 10^{-3}]$ | $[-10^{-3}, 10^{-3}]$ | 0.05 |
| Exploration steps | $28 \cdot 10^3$ | $48 \cdot 10^3$ | $35 \cdot 10^3$ | $28 \cdot 10^3$ | $15 \cdot 10^5$ |
| Batch size | 164 | 32 | 256 | 92 | 32 |
| Target model update | $10^{-3}$ | $10^{-3}$ | $10^{-3}$ | $10^{-2}$ | $10^3$ |
| Initial random steps - Training | 0 | 0 | 0 | 0 | 0 |
| Initial random steps - Testing | 10 | 10 | 10 | 10 | 10 |
| Frame-skip | 4 | 4 | 4 | 4 | 4 |
| Projection - Regularization term | $10^{-6}$ | $10^{-6}$ | $10^{-6}$ | $10^{-6}$ | - |

Table 4: Training settings for the attacker.

