# OpenReview forum: "Optimal Attacks on Reinforcement Learning Policies"
_ICLR.cc/2020/Conference — Reject_

### Official Review · AnonReviewer1 · 2019-10-23
**Official Blind Review #1**

**Rating:** 3

**Review:**

This paper investigates the design of adversarial policies (where the action of the adversarial agent corresponds to a perturbation in the state perceived by the primary agent). In particular it focuses on the problem of learning so-called optimal adversarial policies, using reinforcement learning.

I am perplexed by this paper for a few reasons:
1)	What is the real motivation for this work?  The intro argues “casting optimal attacks is crucial when assessing the robustness of RL policies, since ideally, the agent should learn and apply policies that resist *any* possible attack”.   If the goal is to have agents that are robust to *any* attacks, then they cannot be robust just to so-called optimal attacks.  And so what is really the use of learning so-called optimal attacks?
2)	The notion itself of “optimal” attack is not clear.  The paper does not properly discuss this.  It quickly proposes one possible definition (p.4): “the adversary wishes to minimize the agent’s average cumulative reward”.  This is indeed an interesting setting, and happens to have been studied extensively in game-theoretic multi-agent systems, but the paper does not make much connection with that literature (apart from a brief mention at bottom of p.2 / top of p.3), so it’s not clear what is new here compared to this.   It’s also not discussed whether it would ever be worthwhile considering other notions of optimality for the adversary, and what would be the properties of those.

So overall, while I find the general area of this work to be potentially interesting, the current framing is not well motivated enough, and not sufficiently differentiated from other work in robust MDPs and multi-agent RL to make a strong contribution yet.

More minor comments:
-	P.3: “very different setting where the adversary has a direct impact on the system” => Clarify what are the implications of this in terms of framework, theory, algorithm.
-	P.4: You assume a valid Euclidean distance for the perturbed state.  Is this valid in most MDP benchmarks?  How is this implemented for the domains in the experiments?  What is the action space considered? Do you always assume a continuous action space for the attacker?
-	P.5: “we can simply not maintain distributions over actions” -> Why not?  Given the definition of perturbation, this seems feasible.
-	P.5:  Eqn 4 is defined for a very specific adversarial reward function. Did you consider others?  Is the gradient always easy to derive?
-	P.6: Eqn (5) & (6): What is “R” here?
-	P.7: Figure 1, top right plot. Seems here that the loss is above 0 for small \epsilon.  Is this surprising?  Actually improving the policy?
-	P.7: What happens if you consider even greater \epsilon?  I assume the loss is greater.  But then the perturbation would be more detectable?  How do you think about balancing those 2 requirements of adversarial attacks?  How should we formalize detectability in this setting?
-	Fig.2: Bottom plots are too small to read.
-	Sec.6:  Can you compare to multi-agent baselines, e.g. Morimoto & Doya 2005.
-	P.8: “We also show that Lipschitz policies have desirable robustness properties.” Can you be more specific about where this is shown formally?  Or are you extrapolating from the fact that discrete mountain car suffers more loss than continuous mountain car?  I would suggest making that claim more carefully.



**Experience Assessment:**

I have published in this field for several years.

**Review Assessment: Checking Correctness Of Derivations And Theory:**

I assessed the sensibility of the derivations and theory.

**Review Assessment: Checking Correctness Of Experiments:**

I carefully checked the experiments.

**Review Assessment: Thoroughness In Paper Reading:**

I read the paper thoroughly.

---

> ### Author Response · Authors · 2019-11-08
> **Answers to reviewer#1 - Part 1/3**
>
> Thank you for your comments and questions. We clarify the motivation of our work below. We plan to include these explanations in the paper.
>
> 1. Rev#1: What is the real motivation for this work? If the goal is to have agents that are robust to *any* attacks, then they cannot be robust just to so-called optimal attacks. And so what is really the use of learning so-called optimal attacks?
>
> The primary objective of the paper is to assess the robustness of policies learnt using RL algorithms, and in particular Deep RL algorithms. To this aim, we investigate the possible impact of attacks of a  given amplitude. More precisely, for a given policy $\pi$, we find the optimal attack, defined as the attack that leads to the most important reward impact. Let R be the reward of the policy $\pi$ without attack, and by R_attack the reward obtained under the optimal attack. Computing R_attack is important, because by definition of the optimal attack, the reward obtained under $\pi$ will always be larger than R_attack, under any attack with the given amplitude. The quantity R_attack and the difference R-R_attack define the robustness of the policy $\pi$. Note that the previously proposed attacks on RL policies were designed using heuristics, and were not optimal. Hence these attacks could not be used to really assess the robustness of a policy (there are attacks with stronger impacts).
>
> The ultimate objective of our research is to be able to design RL algorithms that learn policies robust to any attack with given amplitude. This kind of research question has been recently addressed in supervised learning for example in the already influential paper [1] “Towards Deep Learning Models Resistant to Adversarial Attacks”, Madry et al., ICLR 2018, https://arxiv.org/pdf/1706.06083.pdf. Our paper provides the first step towards answering this question in RL. We are looking for a policy $\pi$ that would yield the best average reward even under attack. It is natural to state the problem of designing such a policy $\pi$ as a robust optimization problem as in [1]:
>
> $$(1) \sup_{\pi} \inf_{\phi\in B} V^{\pi o \phi},$$
>
> where $\phi$ is an attack in $B$ (the set of all possible attacks with bounded amplitude), and $V^{\pi o \phi}$ denotes the average reward of the policy $\pi$ under the attack $\phi$ (see the paper for details). Our paper provides for the first time a way of computing $\inf_{\phi\in B} V^{\pi o \phi}$ for a given policy $\pi$, i.e., of characterizing the optimal attack of a given policy. Characterizing this optimal attack is a necessary first step when solving the optimization problem (1). In [1], combining the knowledge optimal attack and Danskin’s theorem leads to an algorithm solving (1). We could do the same here for RL.
>
> 2. Rev#1: The notion itself of “optimal” attack is not clear. […] It quickly proposes one possible definition (p.4): “the adversary wishes to minimize the agent’s average cumulative reward”. This is indeed an interesting setting, and happens to have been studied extensively in game-theoretic multi-agent systems […], so it’s not clear what is new here compared to this. It’s also not discussed whether it would ever be worthwhile considering other notions of optimality for the adversary, and what would be the properties of those.
>
> About “optimal” attacks. We hope that the notion of optimality is now clear from the above discussion (our answer to 1)), and that it is well motivated.
>
> About the connection to game theory. As suggested by the reviewer, we could model the problem as a zero-sum game where the main agent strategy is a policy, and that of the adversary is an attack. There are indeed a lot of related work on zero-sum games. However, the game is actually extremely complex – as for example, the set of possible strategies for the adversary consists of all possible attacks (this is a very large set). To our knowledge, the game considered here does not correspond to anything that you can find in the literature. Actually, what we do in the paper is to compute the ‘best response’ strategy of the attacker to a given strategy (a policy $\pi$) of the main agent. And even this simple game-theoretical task of computing a best response is very hard. We show that this best response (the optimal attack) is the solution of an MDP. In the case of black-box attacks, this solution can be learnt using RL algorithms.About other notions of optimality. In the paper, we mainly focus on characterizing attacks that lead to the lowest reward for the main-agent. However, as mentioned in the paper, we can define the adversary’s reward as we want (see Page 4, where we explain that the adversary may wish to guide the systems towards particular states). The notion of optimal attacks would be then related to the way we define the adversary’s rewards.

---

> ### Author Response · Authors · 2019-11-08
> **Answers to reviewer#1 - Part 2/3**
>
> 3. Rev#1: P.3: “very different setting where the adversary has a direct impact on the system” => Clarify what are the implications of this in terms of framework, theory, algorithm. […] Sec.6: Can you compare to multi-agent baselines, e.g. Morimoto & Doya 2005.
>
> The reviewer refers to the work of Morimoto & Doya[2005]; Pinto et al. [2017]. In these papers, they
> consider an adversary that can directly affect the dynamics of the system. In our case, the adversary can only affect what the main agent perceives, not the system dynamics itself. If the main agent does not make use of a feedback control loop, then, whatever action is taken by the adversary, it won’t affect the dynamics of the system.
>
> 4. Rev.#1: P.4: You assume a valid Euclidean distance for the perturbed state. Is this valid in most MDP benchmarks? How is this implemented for the domains in the experiments? What is the action space considered? Do you always assume a continuous action space for the attacker?
>
> For all the environments (except for the grid world), we use the Euclidian norm as a metric. For these environments, the state space is the Euclidian space of dimension d=2 and 4 (Mountain Car and Cartpole), d=8 (LunarLander), and d= 84x84 (for Pong – the number of pixels in an image). Hence the choice of the Euclidian norm is natural and standard in the literature. As a consequence, in a given state, the possible actions that the attacker can select consist in a Euclidian ball of radius $\epsilon$ around the true state. These balls define the action-space of the attacker, which is indeed continuous. Note that in the case of Pong, the attacker extracts features from the state (an image), to reduce the dimensionality of the state space and hence that of its action space.
> For implementation purposes, the actor-network of the attacker is augmented with a projection layer to ensure that the selected action indeed belongs to the prescribed ball (refer to Appendix A.3 for a detailed description, see Figure 11).
>
> 5. Rev#1: P.5: “we can simply not maintain distributions over actions” -> Why not?
>
> As mentioned above in the answer to P.4, the possible actions of the attacker is a ball in an Euclidian space of dimension d (equal to 2,3, 8 or 84x84). Maintaining a density distribution over such a ball is really difficult, especially when the dimension d grows large. Additionally, we should do maintain such a distribution for all possible states (remember that the state is continuous)! For an image, this is equivalent to maintaining a density distribution for each pixel (for a total of 84x84 distributions).
>
> 6. Rev#1: P.5: Eqn 4 is defined for a very specific adversarial reward function. Did you consider others? Is the gradient always easy to derive?
>
> Equation 4 is the gradient update in the case of a zero-sum game. We have not considered other type of rewards since we believe the zero-sum case to be very important. However, it is not difficult to derive different gradient updates since it is just an application of the chain rule.
>
> 7. Rev#1: P.6: Eqn (5) & (6): What is “R” here?
>
> R is an upper bound on the instantaneous reward (we assume that the latter is bounded).
>
> 8. Rev#1: P.7: Figure 1, top right plot. Seems here that the loss is above 0 for small \epsilon. Is this surprising? Actually improving the policy?
>
> Thank you for pointing out this problem. This is actually normal, because the curves show the deviation of the average reward of the policy under attack compared to the reference value provided by OpenAI Gym documentation Brockman et al. (2016). Now as explained in Page 19 of the appendix, the main agent policies were trained until they were within 5% of this reference value across an average of the last 100 episodes. That explains why we can have positive deviation for small values of $\epsilon$. We are sorry for this confusion, and we will change our curves so that they actually represent the deviation compared to the average reward of the actual policies without attacks.
>
> 9. Rev#1: P.7: What happens if you consider even greater \epsilon? I assume the loss is greater. But then the perturbation would be more detectable? How do you think about balancing those 2 requirements of adversarial attacks? How should we formalize detectability in this setting?
>
> Thank you for this very interesting question. Indeed, ideally, we should define $\epsilon$, the amplitude of the attack as a function of the probability that such an attack could be detected. To our knowledge, there are no work dealing with attack detection in generic MDPs. There a few papers on detecting attacks in linear systems (in the control community, see for example “Generalized chi-squared detector for LTI systems with non-Gaussian noise”, by Hashemi-Ruths, ACC 2019. There, Kalman filter can be used to predict the next state, and in turn to detect an attack. Attack detection in general MDP is an interesting direction for future research.

---

> ### Author Response · Authors · 2019-11-08
> **Answers to reviewer#1 - Part 3/3**
>
> 10. Rev#1: P.8: “We also show that Lipschitz policies have desirable robustness properties.” Can you be more specific about where this is shown formally? Or are you extrapolating from the fact that discrete mountain car suffers more loss than continuous mountain car? I would suggest making that claim more carefully.
>
> As pointed out also by the other reviewers, Proposition 5.1. shows this property. It helps understanding
> how to train resilient agent policies. To test Proposition 5.1, we compare the results obtained when attacking a policy trained using DQN and DDPG. DDPG is known to output smooth policies for continuous systems. And indeed, as shown in Figures 1 and 9, policies trained with DDPG are more robust than other policies. In a revised version of the paper, we plan to further illustrate the result of Proposition 5.1 by computing the smoothness of the various policies that we attack. This smoothness can be quantified by the best constant $L$ (defined in Proposition 5.1).

---

### Official Review · AnonReviewer3 · 2019-10-23
**Official Blind Review #3**

**Rating:** 6

**Review:**

Summary:

The authors of this paper propose a novel adversarial attack for deep reinforcement learning. Different from the classical attacks, e.g., FGSM, they explicitly minimize the reward collect by the agent in a form of Markov decision process. Experiment results demonstrate that the proposed approach can damage the well-performed policy with a much bigger performance drop than gradient-based attacks.

Paper strength:

1.	The paper is well-organized and easy to follow.
2.	Model the adversary of reinforcement learning (RL) system as another MDP and solve it with RL is novel and interesting. The proposed attacking diagram can be devised in a pure black-box setting and also can be incorporated with white-box attacks.
3.	With such a strong attack, the authors derive an upper bound on the impact of attacks and shed light on new research studying the robustness of deep RL approach.

Paper weakness:
1.	The author should give more details about how you use a gradient-based exploration to guide the adversary. From my point of view, I think the black-box attack is more practical and interesting. I would like to see the clearer comparison of optimal attack in a pure black-box setting with gradient-based attacks.
2.	Though FGM is not as efficient as the proposed optimal attack, they are simpler than a learning-based approach. Please describe the details and cost of training the attack agency, e.g., the hyper-parameter, number of training iterations.
3.	While the conclusion of smooth policies is more resilience for adversaries is interesting, I would like to see the evaluation results of such a novel finding.


**Experience Assessment:**

I have published one or two papers in this area.

**Review Assessment: Checking Correctness Of Derivations And Theory:**

I assessed the sensibility of the derivations and theory.

**Review Assessment: Checking Correctness Of Experiments:**

I assessed the sensibility of the experiments.

**Review Assessment: Thoroughness In Paper Reading:**

I read the paper at least twice and used my best judgement in assessing the paper.

---

> ### Author Response · Authors · 2019-11-08
> **Answers to Reviewer #3 comments**
>
> We would like to thank the reviewer for valuable and constructive comments. We also appreciate the
> concise summary of the paper.
>
> 1. Rev#3: The author should give more details about how you use a gradient-based exploration to guide the adversary.  […], I think the black-box attack is more >practical and interesting. I would like to see the clearer comparison of  optimal attack in a pure black-box setting with gradient-based attacks.
>
> We agree that the design of black-box attacks is more practical and interesting, and such a design  constitutes the main contribution of our paper. We did not have much space left to investigate white-box attacks, where the main-agent policy could  be known. Nevertheless, we wanted to add a short discussion on how we could exploit this knowledge  to cast attacks. This is done at the bottom of page 5 (“Gradient-based exploration”) and pages 13-14 in the appendix.
>
> To leverage the knowledge of the main-agent policy, we propose a heuristic, consisting in modifying  the way the attacker explores when learning the optimal attack. More precisely, the proposed gradient-based exploration is a combination of the usual noise used to explore in RL and of a term that is inspired by FGM. The idea is that when exploring, we should to put a higher emphasis in the direction  of the gradient of the loss function. We hope that this leads to a faster learning process. You can find a precise description of this heuristic in pages 13-14 in appendix. Exploring in preferred directions introduces a bias. To deal with this, we make use of a Bernoulli random variable $X_t$ that dictates when to use the usual exploration noise, and the “guided” exploration noise. Finally, observe  that we should follow the direction pointed out by FGM only when this is strictly necessary. To do so,  we consider the gap $max_a \pi(a|s) - \min_a \pi(a|s)$. This gap term multiplies the direction given by the gradient. Intuitively, the lower this gap is, the less impact the optimal action has in that state. This guarantees that we follow the gradient of J mostly when we are in critical states. We have tested this type of exploration on Mountaincar, which is an environment that is usually hard to attack. In Figure 10, we can see a significant increase in the training speed using gradient-based exploration. Given the same number of training episodes, a usual exploration achieves a mean reward of -163, whilst gradient-based exploration achieves -173. FGM achieves -134, and without perturbation, the average score is -100.
>
> 2. Rev.#3: Though FGM is not as efficient as the proposed optimal attack, they are simpler than a >learning-based approach. Please describe the details and cost of training the attack agency, e.g., the hyper-parameter, number of training iterations.
>
> FGM indeed looks simpler to implement. However, FGM requires to compute a gradient (e.g. that of  the Q function w.r.t. the state) each time a control action is selected by the main agent. Computing  such a gradient can be involved, for example in environments with a continuous action space. It may not even be feasible to use FGM in real attack scenarios, since computing this gradient may take an  amount of time greater than the control period. Evaluating FGM actually took a longer time than training and evaluating our attack when the action  space is continuous.  Unlike FGM, our attacks need to be trained before the attack actually takes place.
>
> To train the attack, we use an actor-critic architecture, shown in Figure 11. The actor-network is augmented with a projection layer to ensure a small perturbation. In Table 3, we show the hyper-parameters for the main agent, and in Table 4, the hyper-parameters for the adversary. We used roughly the same number of  training steps to train both the adversary and the main agent, and similar replay memory size.
>
> 3. Rev.#3: While the conclusion of smooth policies is more resilience for adversaries is interesting, I would like to see the evaluation results of such a novel finding
>
> To test Proposition 5.1, we compare the results obtained when attacking a policy trained using DQN
> and DDPG. DDPG is known to output smooth policies for continuous systems. And indeed, as shown In Figures 1 and 9, policies trained with DDPG are more robust than other policies.
> In a revised version of the paper, we plan to further illustrate the result of Proposition 5.1 by computing the smoothness of the various policies that we attack. This smoothness can be quantified by the best constant $L$ (defined in Proposition 5.1).

---

### Official Review · AnonReviewer2 · 2019-10-24
**Official Blind Review #2**

**Rating:** 6

**Review:**

The paper investigates adversarial attacks on learned (fixed) policies. In particular, they devise optimal attacks in the sense that e.g. the agent’s objective should be minimised by the attacker. It is assumed that the attacker can manipulate the observed state of the agent at each time step during testing in a restricted way such that perturbations are small in the state or feature space. The paper derives attacker value functions when the attacker’s goal ist to minimise the original agent’s average cumulative reward and shows how to maximise them using gradient methods. In order to show when attacks are more likely to be successful, a theorem is presented and proved that shows that the resilience of a policy against attacks depends on its smoothness. Experiments show that a) attacks that are trained for optimality w.r.t. to minimising the average reward of the agent outperform a baseline method that only optimises a related criterion. b) Agent policies that are trained with DRQN, yielding a recurrent controller, outperform non recursive ones.

Overall, I think this paper can be a good contribution to the subfield of adversarial attacks on MDP policies. In my view, the derivations and overall structure and results of the paper are sound. However, I would have liked to get a better understanding and motivation for the investigated problem setting, such as projected applications and state or features spaces that could be manipulated in the proposed way.

Pros:
Paper is well written and generally easy to follow.
The derivations look sound and are well motivated. Proposition 5.1. can help in understanding how to train resilient agent policies.
Experiments confirm that optimising for optimal attacks does indeed find empirically better attack strategies.


Cons:
No real application was presented. While applications are imaginable, a real application would have been beneficial. In particular, getting an idea of how real pertubations may look like in a realistic domain. The same holds for the assumption that the state (or features) can be manipulated at each time step, but only slightly.
The implications of Proposition 5.1. were never tested, as far as I understand. You only test DRQN vs DQN, which validates your POMDP assertions, but other comparisons are missing.
The tested Open AI Gym environments are very basic, with the exception of Pong. This reiterates my application argument.

Notation/Writing:
On page 5, last paragraph you should use a different letter for J in order to reduce notational confusion.
Section 6 is not on the same standard as the other sections in terms of writing, e.g. 'We have obtained from 3 policies‘ , some stray ’the’, some plural vs. singular issues.


**Experience Assessment:**

I have read many papers in this area.

**Review Assessment: Checking Correctness Of Derivations And Theory:**

I assessed the sensibility of the derivations and theory.

**Review Assessment: Checking Correctness Of Experiments:**

I assessed the sensibility of the experiments.

**Review Assessment: Thoroughness In Paper Reading:**

I read the paper at least twice and used my best judgement in assessing the paper.

---

> ### Author Response · Authors · 2019-11-08
> **Answers to Rev#2 comments**
>
> We would like to thank the reviewer for their valuable feedback and careful reading. We appreciated the thorough summary that was provided.
>
> 1. REV2: However, I would have liked to get a better understanding and motivation for the investigated problem setting, such as projected applications and state or features spaces that could be manipulated in the proposed way.
>
> One of the motivations is that modern control systems are increasingly relying on advanced infrastructure to get real-time measurements. This increase in complexity increases their exposure to malicious threats. Each measurement signal may be compromised or altered by an adversary. Therefore, resiliency to these attacks is an important property for control systems. Attacks for linear systems are well studied, but little has been done in the field of controlled Markov chains, and that of RL more generally.
>
> 2. REV#2: No real application was presented. While applications are imaginable, a real application would have been beneficial. […].
>
> We agree that a real application would have been beneficial, which is part of the future work. A real application involves the use of reinforcement learning for heat-control in complex systems (like buildings). We would like to verify robustness to adversarial attacks through the methods proposed in this paper. The idea is that an attacker can inject malicious data to your measurement network, or alter the behaviour of one/multiple sensor/s.
>
> 3. REV#2: The implications of Proposition 5.1. were never tested […]
>
> To test Proposition 5.1, we compare the results obtained when attacking a policy trained using DQN and DDPG. DDPG is known to output smooth policies for continuous systems. And indeed, as shown in Figures 1 and 9, policies trained with DDPG are more robust than other policies.
>
> In a revised version of the paper, we plan to further illustrate the result of Proposition 5.1 by computing the smoothness of the various policies that we attack. This smoothness can be quantified by the best constant $L$ (defined in Proposition 5.1).
>
> 4. REV#2: The tested Open AI Gym environments are very basic, with the exception of Pong[…]
>
> We chose environments that may be close to real control systems, such as LunarLander, where the state space is quite rich. As you pointed out, Pong is not a basic environment, but an attacker could use the same methodology to other types of environments and reduce the dimensionality of the state (notice that in the case of Pong we even obtain a small finite state space).

---

### Author Response · Authors · 2019-11-15
**Last day of discussion**

Rev. #2 and #3: We hope that our explanations were clear. Please let us know if you have comments or further questions.

For Rev. #1 we hope that our explanations were insightful, and that the motivation of our work is now clear and interesting. We believe that the main message of our paper was not understood (our fault if this was unclear), but we hope that our clarifications helped. Please let us know if everything is clear now, and if you have further questions.

This is the last day when we can discuss, thank you very much.

Best

---

### Decision · Program_Chairs · 2019-12-19

**Decision:**

Reject

**Comment:**

This paper studies the problem of devising optimal attacks in deep RL to minimize the main agent average reward. In the white-box attack setting, optimal attacks amounts to solving a Markov Decision Process, while in black-box attacks, optimal attacks can be trained using RL techniques. Empirical efficiency of the attacks was demonstrated. It has valuable contributions on studying the adversarial robustness on deep RL. However, the current motivation and setup needs to be made clearer, and so is not being accepted at this time. We hope for these comments to help improve a future version.